# No Remdesivir Resistance Observed in the Phase 3 Severe and Moderate COVID-19 SIMPLE Trials

**DOI:** 10.3390/v16040546

**Published:** 2024-03-31

**Authors:** Charlotte Hedskog, Christoph D. Spinner, Ulrike Protzer, Dieter Hoffmann, Chunkyu Ko, Robert L. Gottlieb, Medhat Askar, Meta Roestenberg, Jutte J. C. de Vries, Ellen C. Carbo, Ross Martin, Jiani Li, Dong Han, Lauren Rodriguez, Aiyappa Parvangada, Jason K. Perry, Ricard Ferrer, Andrés Antón, Cristina Andrés, Vanessa Casares, Huldrych F. Günthard, Michael Huber, Grace A. McComsey, Navid Sadri, Judith A. Aberg, Harm van Bakel, Danielle P. Porter

**Affiliations:** 1Gilead Sciences, Inc., Foster City, CA 94404, USA; ross.martin@gilead.com (R.M.); jiani.li23@gilead.com (J.L.); dong.han1@gilead.com (D.H.); lauren.rodriguez14@gilead.com (L.R.); pc.parvangada@gilead.com (A.P.); jason.perry@gilead.com (J.K.P.); danielle.porter@gilead.com (D.P.P.); 2TUM School of Medicine and Health, Department of Clinical Medicine—Clinical Department for Internal Medicine II, University Medical Center, Technical University of Munich, 81675 Munich, Germany; christoph.spinner@mri.tum.de; 3German Center for Infection Research (DZIF), Munich Partner Site, 81675 Munich, Germany; protzer@tum.de (U.P.); dieter.hoffmann@mri.tum.de (D.H.); 4Institute of Virology, Technical University of Munich School of Medicine, 81675 Munich, Germany; ckko@krict.re.kr; 5Institute of Virology, Helmholtz Munich, 85764 Munich, Germany; 6Infectious Diseases Therapeutic Research Center, Korea Research Institute of Chemical Technology (KRICT), Daejeon 34114, Republic of Korea; 7Center for Advanced Heart and Lung Disease, Department of Internal Medicine, Baylor University Medical Center, Dallas, TX 75246, USA; robert.gottlieb@bswhealth.org (R.L.G.); maskar@qu.edu.qa (M.A.); 8Baylor Scott & White Research Institute, Dallas, TX 75246, USA; 9Department of Internal Medicine, Texas A&M Health Science Center, Dallas, TX 75246, USA; 10Department of Internal Medicine, Burnett School of Medicine at TCU, Fort Worth, TX 76109, USA; 11QU Health and Department of Immunology, College of Medicine, Qatar University, Doha P.O. Box 2713, Qatar; 12Leiden University Medical Center for Infectious Diseases (LUCID), 2333 ZA Leiden, The Netherlands; m.roestenberg@lumc.nl (M.R.); j.j.c.de_vries@lumc.nl (J.J.C.d.V.); e.c.carbo@amsterdamumc.nl (E.C.C.); 13Vall d’Hebron Hospital Universitari, Vall d’Hebron Institut de Recerca (VHIR), Vall d’Hebron Barcelona Hospital Campus, Medicine Department, Universitat Autònoma de Barcelona, 08035 Barcelona, Spain; ricard.ferrer@vallhebron.cat (R.F.); andres.anton@vallhebron.cat (A.A.); cristina.andresverges@vallhebron.cat (C.A.); vanessa.casares@vhir.org (V.C.); 14Department of Infectious Diseases and Hospital Epidemiology, University Hospital Zurich, 8057 Zurich, Switzerland; huldrych.guenthard@usz.ch; 15Institute of Medical Virology, University of Zurich, 8057 Zurich, Switzerland; 16Department of Medicine, University Hospitals of Cleveland and Case Western Reserve University, Cleveland, OH 44106, USA; grace.mccomsey@uhhospitals.org (G.A.M.); navid.sadri@uhhospitals.org (N.S.); 17Department of Medicine, Icahn School of Medicine at Mount Sinai, New York, NY 10029, USA; judith.aberg@mssm.edu; 18Department of Genetics and Genomic Sciences, Icahn School of Medicine at Mount Sinai, New York, NY 10029, USA; harm.vanbakel@mssm.edu

**Keywords:** remdesivir, SARS-CoV-2, resistance, genotyping, phenotyping, Nsp12

## Abstract

Remdesivir (RDV) is a broad-spectrum nucleotide analog prodrug approved for the treatment of COVID-19 in hospitalized and non-hospitalized patients with clinical benefit demonstrated in multiple Phase 3 trials. Here we present SARS-CoV-2 resistance analyses from the Phase 3 SIMPLE clinical studies evaluating RDV in hospitalized participants with severe or moderate COVID-19 disease. The severe and moderate studies enrolled participants with radiologic evidence of pneumonia and a room-air oxygen saturation of ≤94% or >94%, respectively. Virology sample collection was optional in the study protocols. Sequencing and related viral load data were obtained retrospectively from participants at a subset of study sites with local sequencing capabilities (10 of 183 sites) at timepoints with detectable viral load. Among participants with both baseline and post-baseline sequencing data treated with RDV, emergent Nsp12 substitutions were observed in 4 of 19 (21%) participants in the severe study and none of the 2 participants in the moderate study. The following 5 substitutions emerged: T76I, A526V, A554V, E665K, and C697F. The substitutions T76I, A526V, A554V, and C697F had an EC_50_ fold change of ≤1.5 relative to the wildtype reference using a SARS-CoV-2 subgenomic replicon system, indicating no significant change in the susceptibility to RDV. The phenotyping of E665K could not be determined due to a lack of replication. These data reveal no evidence of relevant resistance emergence and further confirm the established efficacy profile of RDV with a high resistance barrier in COVID-19 patients.

## 1. Introduction

Remdesivir (RDV) is a nucleotide analog prodrug that is intracellularly metabolized into an analog of adenosine triphosphate (RDV-TP) that inhibits viral RNA polymerases [1,2]. RDV has broad spectrum activity against coronavirus (SARS-CoV, MERS-CoV, and SARS-CoV-2), filovirus, paramyxovirus, and pneumovirus families [3,4,5,6]. RDV is approved for the treatment of COVID-19 in both hospitalized and non-hospitalized patients, with clinical benefit demonstrated in multiple Phase 3 trials. The efficacy data from the pivotal Phase 3, randomized, double-blind, placebo-controlled study ACTT-1 showed a statistically significant shorter median time to recovery for participants with severe and critical COVID-19 pneumonia treated with RDV compared with those in the placebo arm [7]. In the Phase 3 SIMPLE studies (NCT04292899 and NCT04292730), RDV was evaluated in hospitalized participants with severe or moderate COVID-19 pneumonia, respectively. In participants with severe COVID-19 (pneumonia and room-air oxygen saturation of ≤94%), an analysis of the primary endpoint demonstrated that treatment with RDV for up to 5 days and treatment with RDV for up to 10 days resulted in similar odds of improved clinical status on day 14 [8]. In participants with moderate COVID-19 (pneumonia and room-air oxygen saturation of >94%), the primary endpoint results showed that RDV administered for 5 days resulted in significantly better odds of improvement in clinical status on day 11, as assessed by a 7-point ordinal scale, compared with those who received only standard-of-care treatment [9]. Subsequently, in the Phase 3 randomized, double-blind, placebo-controlled PINETREE study, treatment with RDV for 3 days, administered in an outpatient setting to participants with early-stage COVID-19 who were at risk of disease progression, resulted in a statistically significant reduction of 87% in COVID-19-related hospitalization or all-cause death by day 28 compared with the placebo [10].

The development of resistance to RDV has been investigated in in vitro selection experiments using RDV or its parent nucleoside analog GS-441524. In one study using GS-441524 and SARS-CoV-2, virus pools emerged after 13 passages expressing amino acid substitutions V166A, N198S, S759A, V792I, C799F, and C799R in Nsp12, the SARS-CoV-2 RNA-dependent RNA polymerase (RdRp) and target of RDV. The phenotypic testing of virus from passage 13 resulted in RDV 50% inhibition of virus replication (EC_50_) fold changes between 2.6 and 10 from the SARS-CoV-2 A lineage WA1 reference strain [11]. In a second in vitro selection experiment, using RDV and a SARS-CoV-2 isolate containing the P323L substitution in Nsp12, a single amino acid substitution at V166L emerged after 17 passages. Phenotypic testing of recombinant SARS-CoV-2 viruses containing P323L alone or P323L + V166L in combination resulted in RDV EC_50_ fold changes between 1.2 and 1.5 from the WA1 reference strain [12]. In another study, resistance selection with RDV and SARS-CoV-2 identified a single Nsp12 amino acid substitution, E802D, in two virus populations with a minimal loss of susceptibility to RDV (2.5-fold reduction in susceptibility to RDV) [13].

To investigate possible drug resistance development in patients treated with RDV in the Phase 3 SIMPLE studies, SARS-CoV-2 genotypic and phenotypic analyses were conducted retrospectively from stored virology samples.

## 2. Materials and Methods

### 2.1. SIMPLE Study Designs

Design details for the SIMPLE studies have been previously published [8 (NCT04292899), 9 (NCT04292730)]. Briefly, the two Phase 3, randomized studies evaluated the safety and antiviral activity of RDV in participants with severe and moderate COVID-19, respectively. The studies were conducted early in the pandemic, and participants were enrolled between March and May 2020 (last patient, last visit June 2020 for both studies). The trial protocol was designed such that all core testing could be performed locally at the site; thus, although the study was conducted at high-quality centers, most did not have local sequencing capabilities. All patients or their legally authorized representative provided written informed consent prospectively, including for future testing of samples collected during the study. The trials were approved by the institutional review board or ethics committee at each site and were conducted in compliance with the Declaration of Helsinki Good Clinical Practice guidelines and local regulatory requirements.

The severe COVID-19 study enrolled hospitalized participants with confirmed SARS-CoV-2 infection with radiographic evidence of pulmonary infiltrates and who had either an oxygen saturation < 94% or were receiving supplemental oxygen.

The moderate COVID-19 study enrolled hospitalized participants with confirmed SARS-CoV-2 infection with moderate COVID-19 pneumonia (defined as any radiographic evidence of pulmonary infiltrates and an oxygen saturation > 94% on room air). 

### 2.2. Virology Data Collection

The collection of virology samples by the study sites for resistance testing was optional in the study protocols. To conduct virology analysis, study sites with virology sample collection and sequencing capabilities were identified through a questionnaire. Sequencing and related viral load data were obtained retrospectively from 10 of 174 study sites for Parts A and B of the severe COVID-19 study and from 6 of 139 study sites for Parts A and B of the moderate COVID-19 study. The sequencing analysis included any participant with a virology sample collected with detectable viral load who received at least one dose of RDV or was assigned to standard of care.

For the severe COVID-19 study, the sample types collected included nasal swabs, nasal/oropharyngeal swabs, naso-oral swabs, nasopharyngeal swabs, throat swabs, swabs (not specified), sputum, endotracheal tube aspirate, and bronchoalveolar lavage. For the moderate COVID-19 study, the sample types collected included nasal swabs, naso-oral swabs, nasopharyngeal swabs, oropharyngeal swabs, and swabs (not specified).

### 2.3. Sequencing Methodologies

Sequencing was performed retrospectively from stored upper- and lower-respiratory samples. The SARS-CoV-2 whole genome or Nsp12 was amplified by gene-specific primers or enriched by probes and sequenced by locally available assays using either Sanger sequencing, Illumina MiSeq, Illumina Novaseq 6000, or the Ion Torrent platform. 

The Nsp12 sequencing was conducted as follows: Nsp12 was amplified using conventional nested PCR methodology using Phusion Hot start Flex Master Mix (New England Biolabs, Ipswich, MA, USA). The outer and inner PCR reaction generated products of 1481 and 1048 base pairs, respectively. Population sequencing (Sanger) was performed, and primer sequences were removed prior to data analysis, generating a sequence with length of 1028 nucleotides. 

SARS-CoV-2 whole-genome sequencing was conducted using the following 5 methodologies available at the local study sites: (1) The Twist Bio hybrid capture target enrichment protocol (Twist Bioscience, South San Francisco, CA, USA): Briefly, the nucleic acid was extracted, and cDNA was generated according to the manufacturer’s instructions. Libraries were amplified from the cDNA using a PCR thermal cycler. The SARS-CoV-2 virus sequence was then targeted and enriched during a hybridization reaction with a biotin-attached probe specific to the SARS-CoV-2 virus. The enriched SARS-CoV-2 targeted regions were sequenced by Illumina MiSeq. (2) Illumina Ampliseq SARS-CoV-2 Research panel (Illumina, San Diego, CA, USA) [14]: Briefly, the nucleic acid was extracted, and cDNA and amplicons were generated according to the manufacturer’s instructions. The use of proprietary primer modifications allowed for the removal of the primer sequenced during library preparation. The generated libraries were sequenced using Illumina Novaseq 6000 sequencer. (3) SARS-CoV-2 amplification using ARTIC V3 primer set [15]: Briefly, the nucleic acid was extracted, and cDNA was generated using random hexamers. SARS-CoV-2 was amplified using ARTIC V3 primers with adapter sequences according to the instructions from Illumina COVIDseq Test (Illumina, San Diego, CA, USA), nCoV-2019 sequencing protocol v3 [16], or Qiagen SARS-CoV-2 research panel (Qiagen, Hilden, Germany). The generated libraries were sequenced using Illumina MiSeq (single or paired-end reads). (4) Ion AmpliSeq™ SARS-CoV-2 Research panel (Life Technologies Corporation, Carlsbad, CA, USA): Briefly, the nucleic acid was extracted. cDNA synthesis and the preparation of libraries was performed according to the manufacturer’s instructions using primer pools that target 237 amplicons specific to SARS-CoV-2. The libraries were sequenced using the Ion Torrent platform. (5) SARS-CoV-2 was sequenced using the method described by Gonzalez-Reiche [17]. Briefly, nucleic acid was extracted using QIAamp Viral RNA Minikit (QIAGEN, Cat. No. 52904) per the manufacturer’s instructions. In brief, cDNA synthesis was performed using random hexamers. Sample preparation for sequencing was performed using whole-genome amplification with custom-designed tiling primers generating ~1.5 and ~2 kb amplicons with ~200 bp overlaps between each region and the Artic Consortium protocol v3 (https://artic.network/ncov-2019, accessed on 1 February 2021), with modifications. The libraries were sequenced using Illumina MiSeq.

### 2.4. Sequencing Data Processing

Sequencing data were received from the sites as FASTQ files (single or paired-end) that were split per sample and amplification pool. Internally developed software was used to process and align deep sequencing data. Briefly, sequence reads from FASTQ files were evaluated to remove adapter sequences and trim low-quality phred quality scores using Trimmomatic [18]. Reads were aligned to the host and SARS-CoV-2 reference sequence and reads matching the host were removed. Reads aligned to the SARS-CoV-2 reference sequence (Wuhan-Hu-1, NC_045512.2) using a SMALT aligner [19] were further evaluated. Read ends that overlapped with the amplification primers were clipped from the aligned reads based on genomic coordinates. Variant calls at nucleotide and amino acid levels were rolled up across amplification pools, if relevant, and reported per sample. To report in-frame insertions and deletions (indels), the amino acid realignment of reads with indels was performed. Reads containing frameshift indels were trimmed to exclude the region with an indel from further amino acid variant analysis. All aligned, trimmed reads were then translated in-frame, and amino acid changes from the reference sequence were tabulated. Nucleotide variants were reported in the consensus sequence, and mixture codes were used when more than one base was present at ≥15% of the viral population. Indels were reported in the consensus sequence when present at ≥50% of the viral population. 

### 2.5. SARS-CoV-2 Lineage Determination

SARS-CoV-2 lineage was determined by Pangolin Software v.2.3.8 (pangoLEARN version 2021-04-21, https://github.com/cov-lineages/pangolin, accessed on 21 April 2021) using SARS-CoV-2 whole-genome consensus sequences. A sequence coverage threshold of least 70% of the SARS-CoV-2 coding region was used for lineage determination.

### 2.6. SARS-CoV-2 Sequence Analysis

The primary analysis was focused on the identification of amino acid substitutions in the SARS-CoV-2 Nsp12 RdRp gene. Amino acid substitutions identified in SARS-CoV-2 Nsp8, Nsp10, Nsp13, and Nsp14 genes were investigated as secondary analyses. Any pretreatment sequence (day 1 or earlier) was considered as the baseline sequence. Amino acid substitutions in SARS-CoV-2 Nsp12 compared with the reference sequence (Wuhan-Hu-1, NC_045512.2) that occurred in ≥2 participants at baseline were reported. Post-baseline sequences were compared with participant-specific baseline sequences to determine whether any amino acid substitutions had emerged in the Nsp12 gene during or after treatment.

### 2.7. SARS-CoV-2 Phenotypic Analysis

Phenotypic analysis was attempted on post-baseline clinical isolates with identified treatment-emergent amino acid substitutions compared with their baseline sample in the SARS-CoV-2 Nsp12. Site-directed mutants were generated and tested using a replicon assay. The replicon system and the procedures required to assemble and produce non-infectious SARS-CoV-2 replicon were adapted and modified from Xie [20] and Zhang [21]. Briefly, the four plasmids that encode SARS-CoV-2 genes for the non-structural proteins and the nucleocapsid were prepared. The plasmids were verified by restriction enzyme BsaI digestion and Sanger sequencing to exclude the introduction of any undesired mutations into the plasmids prior to assembly of the full-length (minus structural gene) SARS-CoV-2 replicon DNA (24.2 kb). DNA fragments for ligation by restriction enzyme BsaI digestion of the Maxiprep plasmids were prepared and assembled into full-length SARS-CoV-2 replicon DNA in vitro using a T4 DNA ligase. The full-length ligation product was verified by The LabChip^®^ GX Touch™ (Perkin Elmer, Boston, MA, USA) and purified by AMPure PB magnetic beads. Full-length replicon RNA was generated by in vitro transcription (IVT) and purified with a MegaClear RNA purification Kit. Huh7-1CN cells were resuspended in cold Opti-MEM to 1.125 × 10^7^ cells per mL. Fifteen micrograms of RNA were mixed with cells and immediately electroporated. In a 96-well plate, compounds were dispensed directly into 20 μL of incubation medium as 3-fold serial dilutions. To the diluted compound solutions, 180 μL electroporated cells were added. The plates were then incubated at 37 °C with 5% CO_2_. At 48 h post-electroporation, 10 μL of supernatant was added to fresh 96-well plates containing 10 μL of lysis solution in each well. The luciferase assay 1× solution was prepared, and 100 μL was added to each well. The relative luciferase signals were calculated by normalizing the luciferase signals of the compound-treated groups to that of the DMSO-treated groups (set as 100%). EC_50_ values were calculated using a nonlinear four-parameter variable slope regression model as the concentration at which there was a 50% decrease in the luciferase reporter signal relative to the DMSO vehicle alone (0% virus inhibition) and uninfected control culture (100% virus inhibition). Two experiments were performed with technical triplicates. The fold change values were calculated by dividing the variant mean EC_50_ by the SH01 reference strain mean EC_50_.

## 3. Results

### 3.1. Sequencing Data Collection

The site collection of virology samples was optional in the study protocols. Sequencing and related viral load data were obtained retrospectively from study sites with sample collection and local sequencing capabilities (10 of 183 sites; Figure 1; Appendix A). Sequencing was restricted to samples with positive SARS-CoV-2 PCR with sufficient viral copy number to permit sequencing. Overall, sequencing was obtained from 73 of 4838 (1.5%) participants at baseline and 32 (0.7%) participants post-baseline in the severe COVID-19 study, and 14 of 1087 (1.3%) participants at baseline and 6 (0.6%) participants post-baseline in the moderate COVID-19 study. Among these, 19 (0.4%) participants in the severe COVID-19 study and 4 (0.4%) participants in the moderate COVID-19 study had sequencing data available at both baseline and post-baseline timepoints (Table 1). In the RDV treatment group, a majority of participants received 10 days of RDV treatment.

### 3.2. Lineage and Baseline Nsp12 Substitutions

In both the moderate and severe COVID-19 studies, P323L was the most frequently observed amino acid substitution in SARS-CoV-2 Nsp12 at baseline compared with the reference sequence (Wuhan-Hu-1, NC_045512.2). Among the participants with sequence coverage at position 323, this substitution was observed in 50 of 53 participants (94.3%) in the severe COVID-19 study and 8 of 9 participants (88.9%) in the moderate COVID-19 study (Table 2). The Nsp12 substitution A554V was observed in two participants at baseline in the severe COVID-19 study, and both reverted to wildtype during RDV treatment.

The high prevalence of P323L is consistent with its presence in most circulating SARS-CoV-2 variants, including B.1, B.1.1, B.1.119, and B.1.610, which were the most common lineages in these studies. P323L has been phenotyped using the recombinant SARS-CoV-2 system and does not impact susceptibility to RDV in vitro [22]. No participants with variants of concern or interest (VOC/VOI), as defined by the World Health Organization (WHO) or Centers for Disease Control and Prevention (CDC), were identified in these studies conducted in the early portion of the pandemic (enrollment concluded by the end of May 2020).

### 3.3. Post-Baseline Amino Acid Substitutions in SARS-CoV-2 Nsp12

Both baseline and post-baseline sequencing data were obtained from 19 participants in the severe COVID-19 study, all of whom received RDV, and from 4 participants in the moderate COVID-19 study, 2 of whom received RDV and 2 of whom received the standard of care. Post-baseline substitutions in SARS-CoV-2 Nsp12 were observed in 4 of 19 participants (21.1%) in the severe COVID-19 study and none of the 4 participants in the moderate COVID-19 study. The following five substitutions emerged during or after RDV treatment in the severe COVID-19 study: T76T/I, A526V, A554V, E665K, and C697C/F (Table 3). A554V and E665K were observed on day 13 in the same participant, but in endotracheal tube aspirate and bronchoalveolar lavage, respectively. None of the substitutions were observed in >1 participant. Based on the analysis of a cryo-EM structure of the SARS-CoV-2 polymerase complex with a pre-incorporated RDV-TP [23], T76I and A526V are located on the surface of the Nsp12 protein distant from the polymerase active site and viral RNA (Figure 2). In this structure, A554V, E665K, and C697F are located closer to the active site but have no direct interaction with the RNA or the incoming nucleotide (Figure 3). As measured from the C1′ atom of RDV-TP to the amino acid Cα, A554V is 13.9 Å, C697F is 16.9 Å, and E665K is 18.0 Å. Based on this analysis, none of the substitutions were expected to impact susceptibility to RDV. 

### 3.4. Phenotypic Analysis of Emergent Nsp12 Substitutions

Phenotyping was conducted using SARS-CoV-2 replicons containing site-directed mutants (SDMs) in Huh7-1CN cells. SDMs were generated for the emergent Nsp12 substitutions T76I, A526V, A554V, E665K, and C697F. The substitutions T76I, A526V, A554V, and C697F had EC_50_ fold changes of ≤1.5 (Table 4), suggesting a similar susceptibility to RDV as the wildtype ancestral reference. For E665K, transfection of the mutant replicons was attempted twice; however, phenotypic results could not be generated due to a lack of replication.

### 3.5. Post-Baseline Amino Acid Substitutions Emerging in Other Proteins of the Polymerase Complex

Of the 19 participants with both baseline and post-baseline sequence data, sequencing of Nsp8, Nsp10, Nsp13, and Nsp14 was obtained from 12 (0.2%) of 4838 participants in the severe COVID-19 study and none in the moderate COVID-19 study. Sequencing coverage across Nsp8, Nsp10, Nsp13, and Nsp14 varied per participant and timepoint (Table 5). Of the participants with sequencing coverage, a single amino acid substitution in Nsp8 was observed in 1 of 9 (11%) participants. No amino acid substitutions were observed in Nsp10 in any of the 10 participants with sequence coverage. A single amino acid substitution in Nsp13 was observed in 1 of 10 (10%) participants, and amino acid substitutions in Nsp14 were observed in 2 of 11 (18%) participants. Overall, none of the substitutions occurred in >1 participant or have been associated with resistance to RDV.

## 4. Discussion

The Nsp12 polymerase target of RDV is highly conserved across coronaviruses, with close to 100% identity of the enzyme active site [22]. The low diversity and high genetic stability of the RNA replication complex suggests a minimal global risk of pre-existing SARS-CoV-2 resistance to RDV [24]. With approval globally, RDV has been broadly used in patients with COVID-19. The emergence of drug-resistant strains is of concern during the widespread use of a therapeutic. Here we investigated the possible development of drug resistance in a subset of participants with severe and moderate COVID-19 who were treated with RDV in the SIMPLE clinical studies conducted early in the pandemic.

Across both the SIMPLE severe and moderate studies, P323L was the most frequently observed baseline amino acid substitution in SARS-CoV-2 Nsp12 compared with the reference sequence. This is consistent with the presence of P323L in most circulating SARS-CoV-2 variants, including B.1, B.1.1, B.1.119, and B.1.610, which were the most common lineages observed in the SIMPLE studies. During the COVID-19 pandemic, new SARS-CoV-2 variants have been emerging and circulating around the world [25,26,27]. Changes in the SARS-CoV-2 genome in “variants of concern” are mostly located in the Spike region, which is the primary antigen under selective pressure from the immune system [28]. The Nsp12 target of RDV has been shown to be highly conserved, and only 2 prevalent changes (P323L and G671S) were observed in an analysis of nearly 6 million publicly available variant isolate sequences [22]. RDV retains potency to both P323L and G671S and their combination [22]. In addition, the antiviral activity of RDV to SARS-CoV-2 variants of concern have been tested in vitro, and RDV maintains potent antiviral activity against clinical isolates of SARS-CoV-2 Alpha, Beta, Gamma, Delta, Epsilon, and Omicron subvariants [22,29,30,31,32].

The in vitro selection of RDV-resistant strains of SARS-CoV-2 requires a high number of passages [11,12,13], suggesting a high barrier to the development of RDV resistance. Substitutions identified during the RDV in vitro resistance selection experiments were rare or not detected across >6 million publicly available Nsp12 consensus sequences [11,12,33]. Interestingly, none of the substitutions observed in the in vitro resistance selection experiments were observed in patients with sequencing data who were treated with RDV in the SIMPLE studies. Among the participants with both baseline and post-baseline sequencing data, post-baseline substitutions in SARS-CoV-2 Nsp12 were observed in 4 of 19 participants in the severe COVID-19 study and none of the 2 participants treated with RDV in the moderate COVID-19 study. Five substitutions emerged during or after RDV treatment. Based on the structural analysis, T76I and A526V are located on the surface of the Nsp12 protein distant from the polymerase active site or viral RNA and are therefore unlikely to impact susceptibility to RDV. The other three substitutions, A554V, E665K, and C697F, are located closer to the active site but have no direct interaction with the RNA or the incoming nucleotide. The phenotypic testing of T76I, A526V, A554V, and C697F generated by site-directed mutagenesis in the replicon system resulted in a ≤1.5-fold change in EC_50_ from the reference wildtype replicon, suggesting a similar susceptibility to RDV as the wildtype reference. The phenotyping of E665K could not be determined due to a lack of replication. As no substitutions with reduced susceptibility to RDV were observed in the SIMPLE studies, it suggests that RDV has a high barrier to resistance development in COVID-19 patients. This observation is concordance with the resistance analysis from the ACTT-1 Phase 3 clinical study, where a similar rate of emergent amino acid substitutions in Nsp12 and other proteins of the replication/transcription complex was observed in participants in the RDV and placebo groups. Importantly in the ACTT-1 study, only 2 participants had an emergence of substitutions associated with low-level reduced remdesivir susceptibility (V792I and C799F; ≤3.4-fold change in EC_50_ compared with wildtype and associated with reduced fitness). Notably, clinical recovery was similar between participants with and without emergent Nsp12 substitutions in the RDV arm, suggesting no impact on clinical outcome [34]. Rare cases of the prolonged shedding of SARS-CoV-2 have been described in older or immunocompromised patients [35,36,37]. Fortunately, surveillance reports continue to show a high sequence conservation of Nsp12 [24,38,39], and the prevalence of substitutions associated with reduced susceptibility to remdesivir is exceedingly low [24,33,40,41].

This study has important limitations. The severe and moderate COVID-19 SIMPLE studies were conducted early in the pandemic (March to June 2020), and due to supply shortages during this time (including personal protective equipment, swabs, viral transport media, and the availability of viral load and sequencing assays), virology sample collection was optional in the study protocol with only a few study sites collecting and analyzing samples. In addition, sequencing was restricted to samples with positive SARS-CoV-2 PCR and sufficient viral copy number to permit sequencing. Thus, a limited number of virology samples were available for sequencing and only a subset of participants had sequencing data at both baseline and post-baseline. All laboratory testing, including sequencing, was performed by local laboratories at the study sites, rather than through a central laboratory, due to logistical constraints at the time. 

Taken together, the phenotypic analysis of emergent Nsp12 substitutions observed in participants with sequencing data available in the SIMPLE studies showed no change in RDV susceptibility, supporting a high barrier to RDV resistance in COVID-19 patients. The fortuitous discordance between in vitro escape mutations and their rare emergence in vivo are consistent with the postulate that the acquisition of mutations in the SARS-CoV-2 Nsp12 with reduced susceptibility to RDV incurs a fitness penalty, preserving RDV susceptibility at the population level. Additional studies are ongoing from RDV clinical trials to further characterize the resistance profile of RDV.

## Figures and Tables

**Figure 1 viruses-16-00546-f001:**
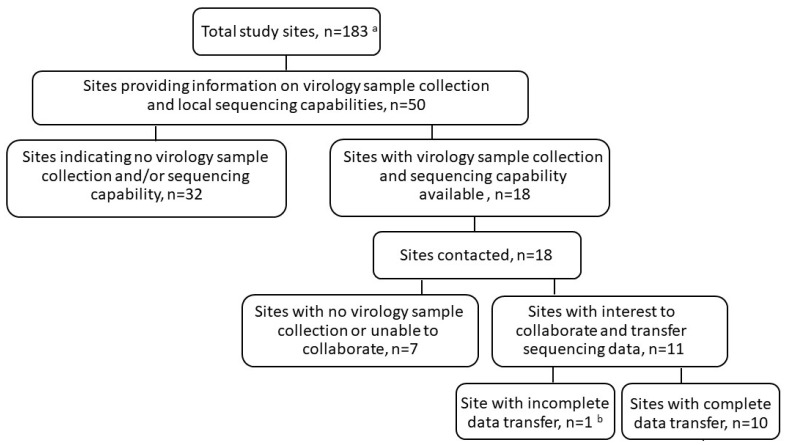
Identification of study sites with virology sample collection and sequencing capabilities. ^a^ The severe and moderate COVID-19 studies comprised 183 sites in total. ^b^ Site unable to provide participant-specific identification numbers to link sequencing data to the corresponding participants in the clinical studies.

**Figure 2 viruses-16-00546-f002:**
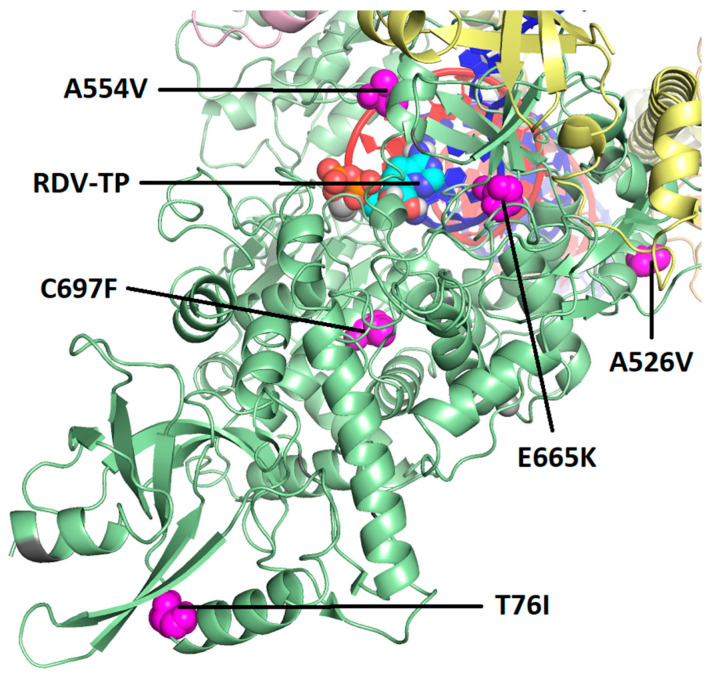
Map of the observed post-baseline amino acid substitutions on the cryo-EM structure of the SARS-CoV-2 polymerase complex with pre-incorporated RDV-TP. RDV-TP = triphosphate form of remdesivir. Pictured is a modified version of the cryo-EM structure, 7UO4 [23], which captures RDV-TP in its pre-incorporated state. The missing primer 3′OH and catalytic metal were added. The full Nsp12 protein is in green, with the locations of the observed post-baseline amino acid substitutions shown in magenta. Pink is Nsp7, and yellow is Nsp8 (two subunits). The template RNA strand is shown in blue and nascent RNA strand in red.

**Figure 3 viruses-16-00546-f003:**
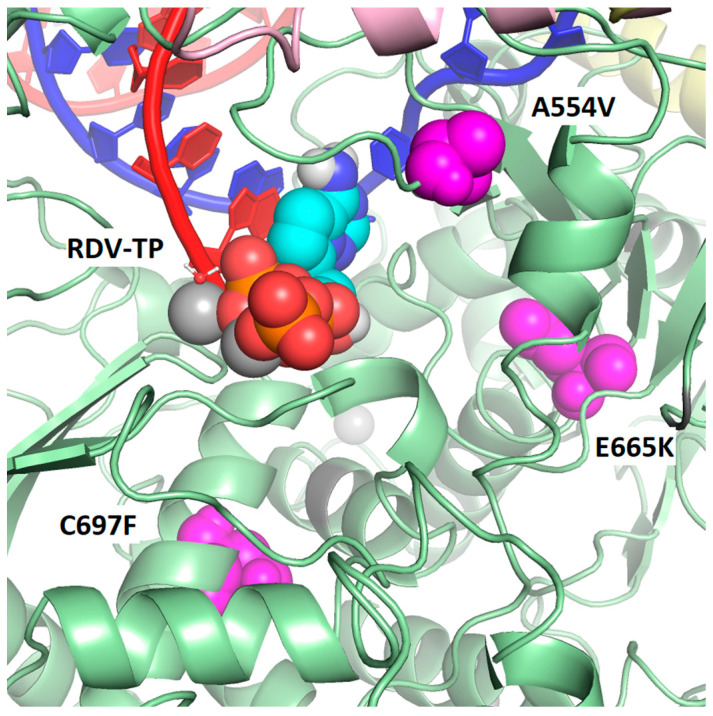
Map of observed post-baseline amino acid substitutions closest to the active site of Nsp12 and on cryo-EM structure of the SARS-CoV-2 polymerase complex with pre-incorporated RDV-TP. None of the observed post-baseline substitutions is in direct contact with RDV-TP or the RNA primer or template strands. However, three of the substitutions are located within 20 Å of the pre-incorporated RDV-TP (as measured from Cα to C1′). A554V is 13.9 Å, C697F is 16.9 Å, and E665K is 18.0 Å. The Nsp12 protein is green with the locations of the substitutions shown in magenta. The template RNA strand is shown in blue and nascent RNA strand in red. Pink is Nsp7, and yellow is Nsp8 (two subunits).

**Table 1 viruses-16-00546-t001:** Summary of SARS-CoV-2 sequencing data from the SIMPLE studies.

	Study Part	Treatment Group	Participants	Participants with Sequencing Data		
				Baseline	Post-Baseline	Both Baseline and Post-Baseline
Severe COVID-19 study ^a^	A	RDV for up to 5 days	200	2	1	0
		RDV for up to 10 days	197	1	1	1
	B	Non-randomized expanded access, RDV for up to 10 days	3597	51	16	9
		Invasive mechanical ventilation, RDV for up to 10 days	844	19	14	9
	Total		4838	73	32	19
Moderate COVID-19 study ^b^	A	RDV for up to 5 days	191	3	0	0
		RDV for up to 10 days	193	0	1	0
		SOC	200	4	2	2
	B	Non-randomized expanded access, RDV for up to 10 days	503	7	3	2
	Total		1087	14	6	4

RDV, remdesivir; SOC, standard of care. ^a^ The Severe COVID-19 study (GS-US-540-5773) was divided into two parts. Part A of the study included 397 participants, and Part B of the study included 4441 participants. All participants in both Parts A and B received RDV treatment (randomized for up to 5 days or up to 10 days in Part A, and non-randomized up to 10 days in Part B; access for patients on invasive mechanical ventilation began upon activation of Part B). The primary efficacy endpoint was clinical status assessed using a 7-point ordinal scale on day 14 of Part A as previously described [8]. ^b^ The Moderate COVID-19 study (GS-US-540-5774) was divided into two parts. Part A of the study included 584 participants, and Part B of the study included 503 participants. Of the 1087 total participants, 200 participants received SOC, and 887 received RDV treatment (5 days or 10 days). The primary efficacy endpoint was clinical status assessed using a 7-point ordinal scale on day 11 of Part A as previously described [9].

**Table 2 viruses-16-00546-t002:** Amino acid substitutions in Nsp12 observed at baseline (compared with Wuhan-Hu-1 reference sequence).

	Number of Participants with Amino Acid Substitutions Occurring in ≥2 Participants at Baseline, n (%) ^a^	
	Severe COVID-19 Study	Moderate COVID-19 Study
P323L	50/53 (94%)	8/9 (88%)
A554V	2/64 (3.1%)	0/13 (0%)

^a^ The denominator is based on the number of participants with sequence coverage at the site of the substitution.

**Table 3 viruses-16-00546-t003:** Amino acid substitutions in Nsp12 detected post-baseline.

	Number of Participants with Emergent Amino Acid Substitutions at Post-Baseline, n (%)									
	Severe COVID-19 Study					Moderate COVID-19 Study				
	Part A		Part B			Part A			Part B	
	RDV for up to 5 Days (n = 0)	RDV for up to 10 Days (n = 1)	Non-randomized Expanded Access (RDV for up to 10 Days) (n = 9)	Invasive Mechanical Ventilation (RDV for up to 10 Days) (n = 9)	Total (n = 19)	RDV for up to 5 Days (n = 0)	RDV for up to 10 Days (n = 0)	SOC (n = 2)	Non-Randomized Expanded Access (RDV for up to 10 Days) (n = 2)	Total in RDV Arms (n = 2)
T76T/I	NA	0	1/9 (11%)	0	1/19 (5%)	NA	NA	0	0	0
A526V	NA	0	0	1/9 (11%)	1/19 (5%)	NA	NA	0	0	0
A554V ^a^	NA	0	0	1/9 (11%) ^a^	1/19 (5%)	NA	NA	0	0	0
E665K ^a^	NA	0	0	1/9 (11%) ^a^	1/19 (5%)	NA	NA	0	0	0
C697C/F	NA	0	1/9 (11%)	0	1/19 (5%)	NA	NA	0	0	0

NA = not applicable; RDV = remdesivir; SOC = standard of care. ^a^ A554V and E665K were observed on day 13 in the same participant, but in endotracheal tube aspirate and bronchoalveolar lavage, respectively.

**Table 4 viruses-16-00546-t004:** RDV EC_50_ against site-directed mutants.

Substitution in Nsp12	RDV EC_50_ (nM)					EC_50_ Fold Change ± SD from WA1
	1st Replicate	2nd Replicate	3rd Replicate	4th Replicate	Mean ± SD	
Wildtype (SH01)	8.83	8.44	9.51	13.03	9.95 ± 2.10	1.00
T76I	-	-	7.16	9.48	8.32 ± 1.63	0.84
A526V	10.69	13.08	-	-	11.89 ± 1.69	1.19
A554V	14.67	14.67	-	-	14.45 ± 0.31	1.45
E665K	No replication				NA	NA
C697F	10.33	9.43	-	-	9.88 ± 0.64	0.99

EC_50_ = half-maximal effective concentration; NA = not applicable; RDV = remdesivir; SD = standard deviation; SH01 = wildtype reference SARS-CoV-2 replicon generated from clinical isolate from Shanghai (lineage B).

**Table 5 viruses-16-00546-t005:** Amino acid substitutions emerging in Nsp8, Nsp10, Nsp13, and Nsp14 at post-baseline in the severe COVID-19 study.

Change from Baseline	Number of Participants with Baseline and Post-Baseline Sequencing Data				
	A		B		Total (n = 12)
	RDV for up to 5 Days (n = 0)	RDV for up to 10 Days (n = 0)	Non-Randomized Expanded Access (RDV for up to 10 Days) (n = 6) ^a^	Invasive Mechanical Ventilation (RDV for up to 10 Days) (n = 6)	
Nsp8 A89T	NA	NA	0	1/6 (17%)	1/9 (11%)
Nsp13 F581L	NA	NA	1/4 (25%)	0	1/10 (10%)
Nsp14 T21R	NA	NA	0	1/6 (17%)	1/11 (9%)
Nsp14 I101V + A287V	NA	NA	1/5 (20%)	0	1/11 (9%)

NA = not applicable; RDV = remdesivir. ^a^ The denominator is based on sequencing coverage for Nsp8, Nsp10, Nsp13, and Nsp14 at baseline and post-baseline and could vary in the different regions of the polymerase complex.

## Data Availability

The data presented in this study are available on request from the corresponding author.

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
