# Peer review of "No Remdesivir Resistance Observed in the Phase 3 Severe and Moderate COVID-19 SIMPLE Trials"

_viruses, 2024, doi:10.3390/v16040546_

Round 1

Reviewer 1 Report

Comments and Suggestions for Authors

In this manuscript, Hedskog et al. performed SARS-CoV-2 resistance analysis by sequencing SARS-CoV-2 genomes in COVID patients treated with RDV, and found 5 substitutions in NSP12 regions. With a SARS-CoV-2 subgenomic replicon system, the authors show that none of them affect susceptibility to RDV, demonstrating no evidence of relevant resistance emergence by RDV treatment. This study provides very interesting and important information regarding drug resistance of COVID patients. I recommend that Tables can be relocated (Tables 1, 3, and 5 are carried over to the next page), but the manuscript can be accepted in a present form.

Author Response

We thank the reviewer for the careful review of this manuscript and have formatted the tables accordingly. 

Reviewer 2 Report

Comments and Suggestions for Authors

Dear authors

After careful evaluation of your manuscript entitled "Viral resistant analyses from the Remdesivir Phase 3 severe 2 and moderateCOVID-19 SIMPLE Trials", which investigates the potential development of viral resistance in patients with moderate and severe COVID-19 infection  treated with Remdesivir, I think that your article is very interesting and could be published without major revisions.The study's design and methodology is appropriate and although the sample size is small and the samples date back to 2020, it is understandable, since it is impossible to perform such extensive analyses as virus sequencing and drug susceptibility, especially before and after treatment, in larger patient numbers. I find the study of great interest and significance to the readership of multiple subspecialties of internal medicine and I think that it further confirms the efficacy of remdesivir that we all are experiencing in everyday clinical practice.

I have only one suggestion for improvement

Could you please revise the title, since it is not very attractive and it is alittle bit confusing

Kind regards

Smaragdi Marinaki

Author Response

We thank the reviewer for the careful review of this manuscript. The current title of the manuscript describes the purpose of this study which is to investigate potential emergence of viral resistance in patients treated with remdesivir in the SIMPLE COVID-19 trials. To address the reviewer’s comments on the title, we have changed it to the following: No remdesivir resistance observed in the phase 3 severe and moderate COVID-19 SIMPLE trials 

Reviewer 3 Report

Comments and Suggestions for Authors

The paper describes the genotypic and phenotypic data of patients enrolled in a study of the treatment of COVID with Remdesivir during the first wave of the epidemic.

Despite the fact that only a small subpopulation of trial participants was evaluated, and with different sequencing techniques, these data are interesting given the low number of publications describing mutations emerging in vivo in subjects treated with Remdesivir, so now. This study confirm that Remdesivir has a high genetic barrier: emerging polymerase mutations are rare, different from one patient to another and do not confer significant phenotypic resistance to Remdesivir.

The study is well done and well presented.

Perhaps the only point worth mentioning may be the tables: they are numerous and very detailed and do not help the reader to vizualise the data. They could be condensed to the main results, or even grouped together for similar topics.

For example, in table 2, corresponding to the mutations found in patients before treatment, it is not necessary to separate patients for whom a different regimen will subsequently be administered. A separation between severe and moderate COVID would be sufficient.

Another example is table 5 corresponding to mutations detected in proteins other than polymerase. There are few patients and few mutations. Having so many rows and columns adds nothing. If a mutation appears in one subject, this intuitively means that it does not occur in others. Lines with only the genes, the mutation detected, and in which patient would be sufficient.   

Author Response

We thank the reviewer for the careful review of this manuscript. We agree with the reviewer and have simplified table 2, table 3 and table 5 accordingly.

Reviewer 4 Report

Comments and Suggestions for Authors

In the article titled “Viral Resistance Analyses from the Remdesivir Phase 3 Severe 2 and Moderate COVID-19 SIMPLE Trials” by Charlotte Hedskog et.al., the authors have presented SARS-CoV-2 resistance analyses from the Phase 3 SIMPLE clinical studies evaluating RDV in hospitalized participants with severe or moderate COVID-19 disease.

 This reviewer would like the authors to address the following point.

 Table 4 – The authors did not observe any significant change in RDV EC50 values while comparing the wildtype(SH01) with three other substitutions A554V, E665K, and C697F which are located closer to the active site. Please provide plausible reasons.

Comments on the Quality of English Language

The english was ok throughout the article.

Author Response

We thank the reviewer for the careful review of this manuscript. Of the substitutions observed, A554V, E665K, and C697F were closer to the active site compared to the other substitutions based on protein structural analysis. These substitutions were located ≥13.9 Å from the active nucleoside triphosphate of RDV (RDV-TP). None of the substitutions observed had any direct interaction with the RNA or incoming nucleotide and were not expected to impact susceptibility to RDV. To clarify this, we updated line 338-339 in the manuscript to describe that none of the substitutions were expected to impact susceptibility to RDV. This is in concordance with the phenotypic data obtained showing that none of the substitutions conferred any reduction in susceptibility to RDV.

Reviewer 5 Report

Comments and Suggestions for Authors

The article describes the results from simple trials, where the most recurrent mutations of RNA polymerase of SARSCoV-2 are dealt with, enlightening the preserved inhibitory activity of the drug remdesivir against wild-type as well as evolutionary mutated virus strains. The Authors assessed the susceptibility of the isolate virus mutants to remdesivir by performing in vitro screening.

The article is well organized and written. The results are interesting and supported by experimental tests.

The article could benefit from an appropriate comparative analysis with data coming from the following relevant papers which have considered the mutational character of SARSCoV2 RNA polymerase and the inhibitory activity of remdesivir. This analysis will allow to better point out what information are new with respect to previous findings. This will strenghten the main concepts and the positive  outcome from this work.

1_Majchrzak M, Madej Ł, Łysek-Gładysińska M, Zarębska-Michaluk D, Zegadło K, Dziuba A, Nogal-Nowak K, Kondziołka W, Sufin I, Myszona-Tarnowska M, Jaśkowski M, Kędzierski M, Maciukajć J, Matykiewicz J, Głuszek S, Adamus-Białek W. The RdRp genotyping of SARS-CoV-2 isolated from patients with different clinical spectrum of COVID-19. BMC Infect Dis. 2024 Mar 4;24(1):281. doi: 10.1186/s12879-024-09146-x. 

2_Pozzi C, Vanet A, Francesconi V, Tagliazucchi L, Tassone G, Venturelli A, Spyrakis F, Mazzorana M, Costi MP, Tonelli M. Antitarget, Anti-SARS-CoV-2 Leads, Drugs, and the Drug Discovery-Genetics Alliance Perspective. J Med Chem. 2023 Mar 23;66(6):3664-3702. doi: 10.1021/acs.jmedchem.2c01229.

3_Delgado S, Somovilla P, Ferrer-Orta C, Martínez-González B, Vázquez-Monteagudo S, Muñoz-Flores J, Soria ME, García-Crespo C, de Ávila AI, Durán-Pastor A, Gadea I, López-Galíndez C, Moran F, Lorenzo-Redondo R, Verdaguer N, Perales C, Domingo E. Incipient functional SARS-CoV-2 diversification identified through neural network haplotype maps. Proc Natl Acad Sci U S A. 2024 Mar 5;121(10):e2317851121. doi: 10.1073/pnas.2317851121. 

4_Lombardo D, Musolino C, Chines V, Caminiti G, Palermo C, Cacciola I, Raffa G, Pollicino T. Assessing Genomic Mutations in SARS-CoV-2: Potential Resistance to Antiviral Drugs in Viral Populations from Untreated COVID-19 Patients. Microorganisms. 2023 Dec 19;12(1):2. doi: 10.3390/microorganisms12010002.

Comments on the Quality of English Language

The English style is good and clear.

Author Response

We thank the reviewer for the careful review of this manuscript. This manuscript focuses on investigation of emergent amino acid substitutions as those could potentially impact the susceptibility to RDV. None of the amino acid substitutions observed in this study conferred reduced susceptibility to RDV. We agree with the reviewer of the importance of studying naturally occurring variants and their susceptibility to antiviral drugs. In the discussion section, we describe susceptibility of RDV to SARS-CoV-2 variants including the common amino acid substitutions P323L and G671S. No change in susceptibility has been observed to any SARS-CoV-2 variant tested to date (line 412-426). Given the limited number of sequences obtained in this study, determination of neuron network maps or investigation of synonymous nucleotide changes with disease severity were not evaluated as that is outside of the scope of the current manuscript.